# Image Synthesis with a Single (Robust) Classifier

**Shibani Santurkar**[*]
MIT
shibani@mit.edu

**Dimitris Tsipras**[*]
MIT
tsipras@mit.edu

**Brandon Tran**[*]
MIT
btran115@mit.edu

**Andrew Ilyas**[*]
MIT
ailyas@mit.edu

**Logan Engstrom**[*]
MIT
engstrom@mit.edu

**Aleksander Mądry**
MIT
madry@mit.edu

## Abstract

We show that the basic classification framework alone can be used to tackle some of the most challenging tasks in image synthesis. In contrast to other state-of-the-art approaches, the toolkit we develop is rather minimal: it uses a *single, off-the-shelf* classifier for *all* these tasks. The crux of our approach is that we train this classifier to be *adversarially robust*. It turns out that adversarial robustness is precisely what we need to directly manipulate salient features of the input. Overall, our findings demonstrate the utility of robustness in the broader machine learning context.[2]

## 1 Introduction

Deep learning has revolutionized the way we tackle computer vision problems. This revolution started with progress on image classification [KSH12; He+15; He+16], which then triggered the expansion of the deep learning paradigm to encompass more sophisticated tasks such as image generation [Kar+18; BDS19] and image-to-image translation [Iso+17; Zhu+17]. Much of this expansion was predicated on developing complex, task-specific techniques, often rooted in the generative adversarial network (GAN) framework [Goo+14]. However, *is there a simpler toolkit for solving these tasks*?

In this work, we demonstrate that basic classification tools *alone* suffice to tackle various image synthesis tasks. These tasks include (cf. Figure 1): *generation* (Section 3.1), *inpainting* (Section 3.2), *image-to-image translation* (Section 3.3), *super-resolution* (Section 3.4), and *interactive image manipulation* (Section 3.5).

Our entire toolkit is based on *a single classifier* (per dataset) and involves performing a simple input manipulation: maximizing predicted class scores with gradient descent. Our approach is thus general purpose and simple to implement and train, while also requiring minimal tuning. To highlight the potential of the core methodology itself, we intentionally employ a generic classification setup (ResNet-50 [He+16] with default hyperparameters) without any additional optimizations (e.g., domain-specific priors or regularizers). Moreover, to emphasize the consistency of our approach, throughout this work we demonstrate performance on *randomly selected* examples from the test set.

The key ingredient of our method is *adversarially robust* classifiers. Previously, Tsipras et al. [Tsi+19] observed that maximizing the loss of robust models over the input leads to realistic instances of other classes. Here we are able to fully leverage this connection to build a versatile toolkit for image synthesis. Our findings thus establish robust classifiers as a powerful primitive for semantic image manipulation, despite them being trained *solely* to perform image classification.

---

[*]Equal contribution

[2]Code and models for our experiments can be found at `https://git.io/robust-apps`.

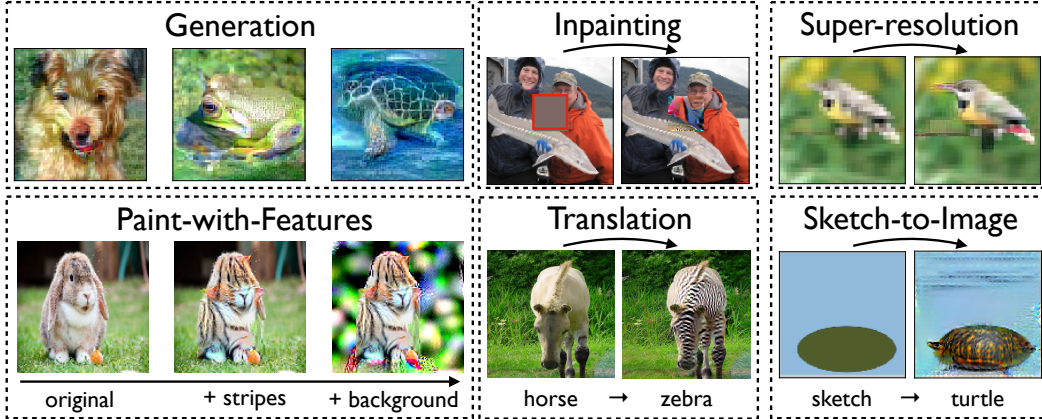

Figure 1: Image synthesis and manipulation tasks performed using a *single* (robustly trained) classifier.

## 2   Robust Models as a Tool for Input Manipulation

Recently, Tsipras et al. [Tsi+19] observed that optimizing an image to cause a misclassification in an (adversarially) robust classifier introduces salient characteristics of the incorrect class. This property is unique to *robust* classifiers: standard models (trained with empirical risk minimization (ERM)) are inherently brittle, and their predictions are sensitive even to imperceptible changes in the input [Sze+14].

Adversarially robust classifiers are trained using the *robust optimization* objective [Wal45; Mad+18], where instead of minimizing the expected loss $\mathcal{L}$ over the data

$$\mathbb{E}_{(x,y)\sim\mathcal{D}}\left[\mathcal{L}(x,y)\right], \tag{1}$$

we minimize the worst case loss over a specific perturbation set $\Delta$

$$\mathbb{E}_{(x,y)\sim\mathcal{D}}\left[\max_{\delta\in\Delta}\ \mathcal{L}(x+\delta,y)\right]. \tag{2}$$

Typically, the set $\Delta$ captures imperceptible changes (e.g., small $\ell_2$ perturbations), and given such a $\Delta$, the problem in (2) can be solved using adversarial training [GSS15; Mad+18].

From one perspective, we can view robust optimization as encoding priors into the model, preventing it from relying on imperceptible features of the input [Eng+19]. Indeed, the findings of Tsipras et al. [Tsi+19] are aligned with this viewpoint—by encouraging the model to be invariant to small perturbations, robust training ensures that changes in the model's predictions correspond to salient input changes.

In fact, it turns out that this phenomenon also emerges when we maximize the probability of a *specific class* (targeted attacks) for a robust model—see Figure 2 for an illustration. This indicates that robust models exhibit more human-aligned gradients, and, more importantly, that we can precisely control features in the input just by performing gradient descent on the model output. Previously, performing such manipulations has only been possible with more complex and task-specific techniques [MOT15; RMC16; Iso+17; Zhu+17]. In the rest of this work, we demonstrate that this property of robust models is sufficient to attain good performance on a diverse set of image synthesis tasks.

## 3   Leveraging Robust Models for Computer Vision Tasks

Deep learning-based methods have recently made significant progress on image synthesis and manipulation tasks, typically by training specifically-crafted models in the GAN framework [Goo+14; ISI17; Zhu+17; Yu+18; BDS19], using priors obtained from deep generative models [Ngu+16;

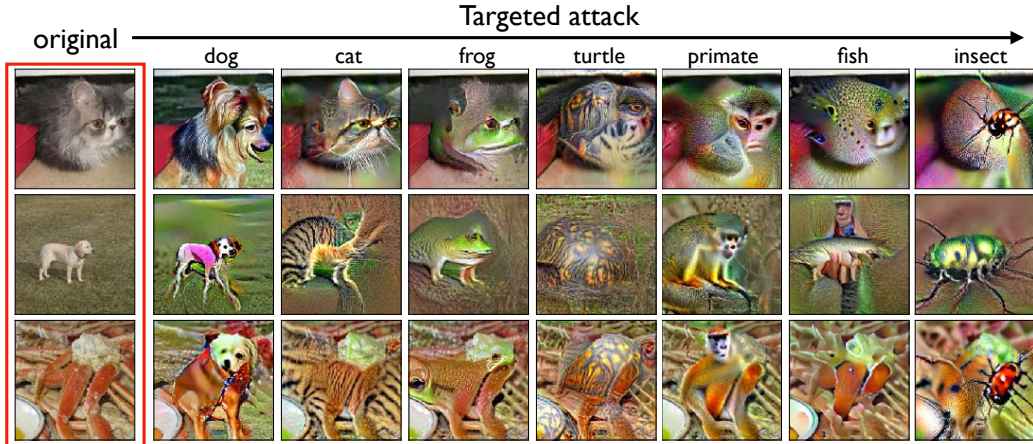

Figure 2: Maximizing class scores of a robustly trained classifier. For each original image, we visualize the result of performing targeted projected gradient descent (PGD) toward different classes. The resulting images actually resemble samples of the target class.

Ngu+17; UVL17; Yeh+17], or leveraging standard classifiers via sophisticated, task-specific methods [MOT15; Oyg15; Tyk16; GEB16]. We discuss additional related work in the following subsections as necessary.

In this section, we outline our methods and results for obtaining competitive performance on these tasks using only robust (feed-forward) classifiers. Our approach is remarkably simple: all the applications are performed using gradient ascent on class scores derived from the same robustly trained classifier. In particular, it does not involve fine-grained tuning (see Appendix A.4), highlighting the potential of robust classifiers as a versatile primitive for sophisticated vision tasks.

## 3.1 Realistic Image Generation

Synthesizing realistic samples for natural data domains (such as images) has been a long standing challenge in computer vision. Given a set of example inputs, we would like to learn a model that can produce novel perceptually-plausible inputs. The development of deep learning-based methods such as autoregressive models [HS97; Gra13; VKK16], auto-encoders [Vin+10; KW15] and flow-based models [DKB14; RM15; DSB17; KD18] has led to significant progress in this domain. More recently, advancements in generative adversarial networks (GANs) [Goo+14] have made it possible to generate high-quality images for challenging datasets [Zha+18; Kar+18; BDS19]. Many of these methods, however, can be tricky to train and properly tune. They are also fairly computationally intensive, and often require fine-grained performance optimizations.

In contrast, we demonstrate that robust classifiers, without any special training or auxiliary networks, can be a powerful tool for synthesizing realistic natural images. At a high level, our generation procedure is based on maximizing the class score of the desired class using a robust model. The purpose of this maximization is to add relevant and semantically meaningful features of that class to a given input image. This approach has been previously used on standard models to perform class visualization—synthesizing prototypical inputs of each class—in combination with domain-specific input priors (either hand-crafted [NYC15] and learned [Ngu+16; Ngu+17]) or regularizers [SVZ13; MOT15; Oyg15; Tyk16].

As the process of class score maximization is deterministic, generating a diverse set of samples requires a random seed as the starting point of the maximization process. Formally, to generate a sample of class $y$, we sample a seed and minimize the loss $\mathcal{L}$ of label $y$

$$x = \underset{\|x'-x_0\|_2 \leq \varepsilon}{\arg\min} \ \mathcal{L}(x', y), \qquad x_0 \sim \mathcal{G}_y,$$

for some class-conditional seed distribution $\mathcal{G}_y$, using projected gradient descent (PGD) (experimental details can be found in Appendix A). Ideally, samples from $\mathcal{G}_y$ should be diverse and statistically similar to the data distribution. Here, we use a simple (but already sufficient) choice for $\mathcal{G}_y$—a

multivariate normal distribution fit to the empirical class-conditional distribution

$$\mathcal{G}_y := \mathcal{N}(\boldsymbol{\mu}_y, \boldsymbol{\Sigma}_y), \quad \text{where } \boldsymbol{\mu}_y = \mathbb{E}_{x \sim \mathcal{D}_y}[x], \ \boldsymbol{\Sigma} = \mathbb{E}_{x \sim \mathcal{D}_y}[(x - \boldsymbol{\mu}_y)^\top (x - \boldsymbol{\mu}_y)],$$

and $\mathcal{D}_y$ is the distribution of natural inputs conditioned on the label $y$. We visualize example seeds from these multivariate Gaussians in Figure 17.

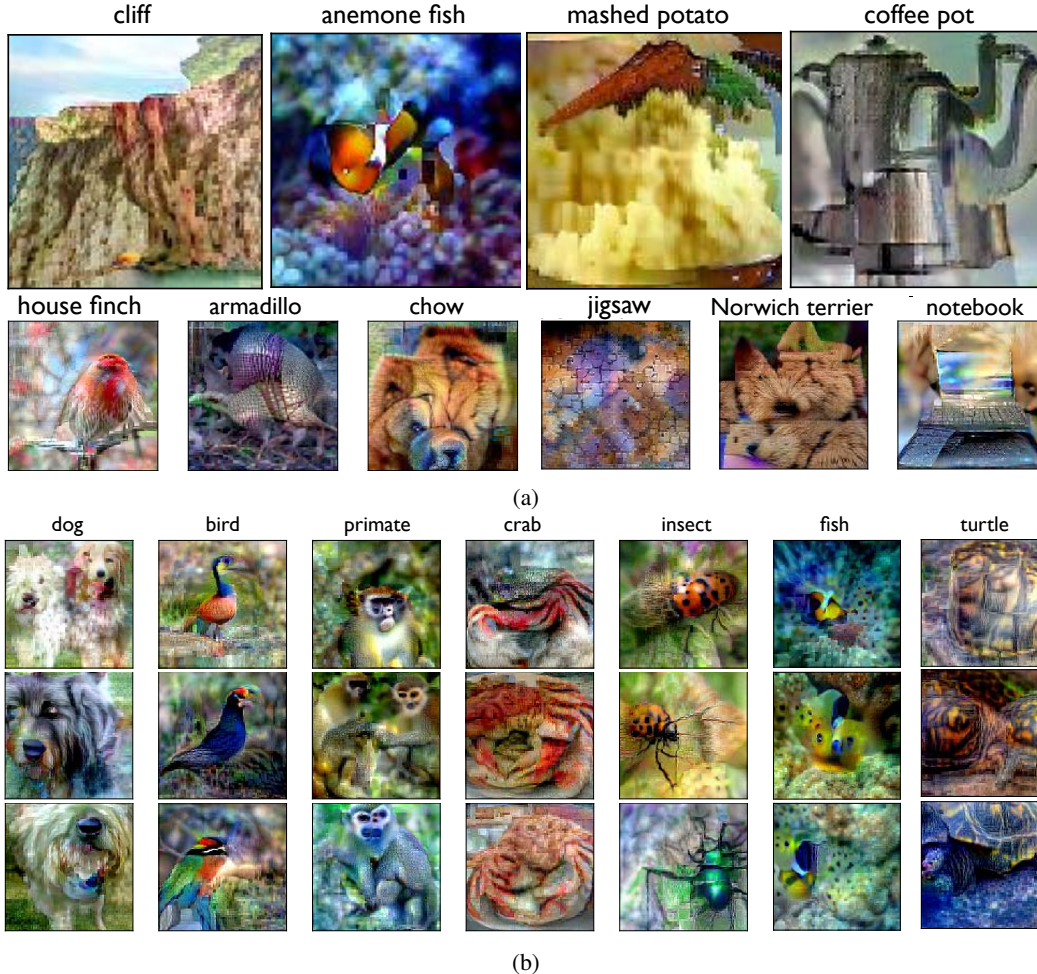

Figure 3: *Random* samples (of resolution 224×224) produced using a robustly trained classifier. We show: (a) samples from several (random) classes of the ImageNet dataset and (b) multiple samples from a few random classes of the restricted ImageNet dataset (to illustrate diversity). See Figures 13, 14, 15, and 16 of Appendix B for additional samples.

This approach enables us to perform *conditional* image synthesis given any target class. Samples (at resolution 224×224) produced by our method are shown in Figure 3 (also see Appendix B). The resulting images are diverse and realistic, despite the fact that they are generated using targeted PGD on off-the-shelf robust models without any additional optimizations. [3]

**Different seed distributions.** It is worth noting that there is significant room for improvement in designing the distribution $\mathcal{G}_y$. One way to synthesize better samples would be to use a richer distribution—for instance, mixtures of Gaussians per class to better capture multiple data modes. Also, in contrast to many existing approaches, we are not limited to a single seed distribution, and we could even utilize other methods (such as procedural generation) to customize seeds with specific structure or color, and then maximize class scores to produce realistic samples (e.g., see Section 3.5).

**Evaluating Sample Quality.** Inception Score (IS) [Sal+16] is a popular metric for evaluating the quality of generated image data. Table 1 presents the IS of samples generated using a robust classifier.

| Dataset | Train Data | BigGAN [BDS19] | WGAN-GP [Gul+17] | Our approach |
|---------|------------|----------------|------------------|--------------|
| CIFAR-10 | $11.2 \pm 0.2$ | 9.22 | $8.4 \pm 0.1$ | $7.5 \pm 0.1$ |
| ImageNet[4] | $331.9 \pm 4.9$ | $233.1 \pm 1$ | 11.6 | $259.0 \pm 4$ |

Table 1: Inception Scores (IS) for samples generated using robustly trained classifiers compared to state-of-the-art generation approaches [Gul+17; SSA18; BDS19] (cf. Appendix A.7.1 for details).

We find that our approach improves over state-of-the-art (BigGAN [BDS19]) in terms of Inception Score on the ImageNet dataset, yet, at the same time, the Fréchet Inception Distance (FID) [Heu+17] is worse (36.0 versus 7.4). These results can be explained by the fact that, on one hand, our samples are essentially adversarial examples (which are known to transfer across models [Sze+14]) and thus are likely to induce highly confident predictions that IS is designed to pick up. On the other hand, GANs are explicitly trained to produce samples that are indistinguishable from true data with respect to a discriminator, and hence are likely to have a better (lower) FID.

## 3.2 Inpainting

Image inpainting is the task of recovering images with large corrupted regions [EL99; Ber+00; HE07]. Given an image $x$, corrupted in a region corresponding to a binary mask $m \in \{0, 1\}^d$, the goal of inpainting is to recover the missing pixels in a manner that is perceptually plausible with respect to the rest of the image. We find that simple feed-forward classifiers, when robustly trained, can be a powerful tool for such image reconstruction tasks.

From our perspective, the goal is to use robust models to restore missing features of the image. To this end, we will optimize the image to maximize the score of the underlying true class, while also forcing it to be consistent with the original in the uncorrupted regions. Concretely, given a robust classifier trained on uncorrupted data, and a corrupted image $x$ with label $y$, we solve

$$x_I = \arg\min_{x'} \mathcal{L}(x', y) + \lambda ||(x - x') \odot (1 - m)||_2 \qquad (3)$$

where $\mathcal{L}$ is the cross-entropy loss, $\odot$ denotes element-wise multiplication, and $\lambda$ is an appropriately chosen constant. Note that while we require knowing the underlying label $y$ for the input, it can typically be accurately predicted by the classifier itself given the corrupted image.

In Figure 4, we show sample reconstructions obtained by optimizing (3) using PGD (cf. Appendix A for details). We can observe that these reconstructions look remarkably similar to the uncorrupted images in terms of semantic content. Interestingly, even when this approach fails (reconstructions differ from the original), the resulting images do tend to be perceptually plausible to a human, as shown in Appendix Figure 12.

## 3.3 Image-to-Image Translation

As discussed in Section 2, robust models provide a mechanism for transforming inputs between classes. In computer vision literature, this would be an instance of *image-to-image translation*, where the goal is to translate an image from a source to a target domain in a semantic manner [Her+01].

In this section, we demonstrate that robust classifiers give rise to a new methodology for performing such image-to-image translations. The key is to (robustly) train a classifier to distinguish between the source and target domain. Conceptually, such a classifier will extract salient characteristics of each domain in order to make accurate predictions. We can then translate an input from the source domain by directly maximizing the predicted score of the target domain.

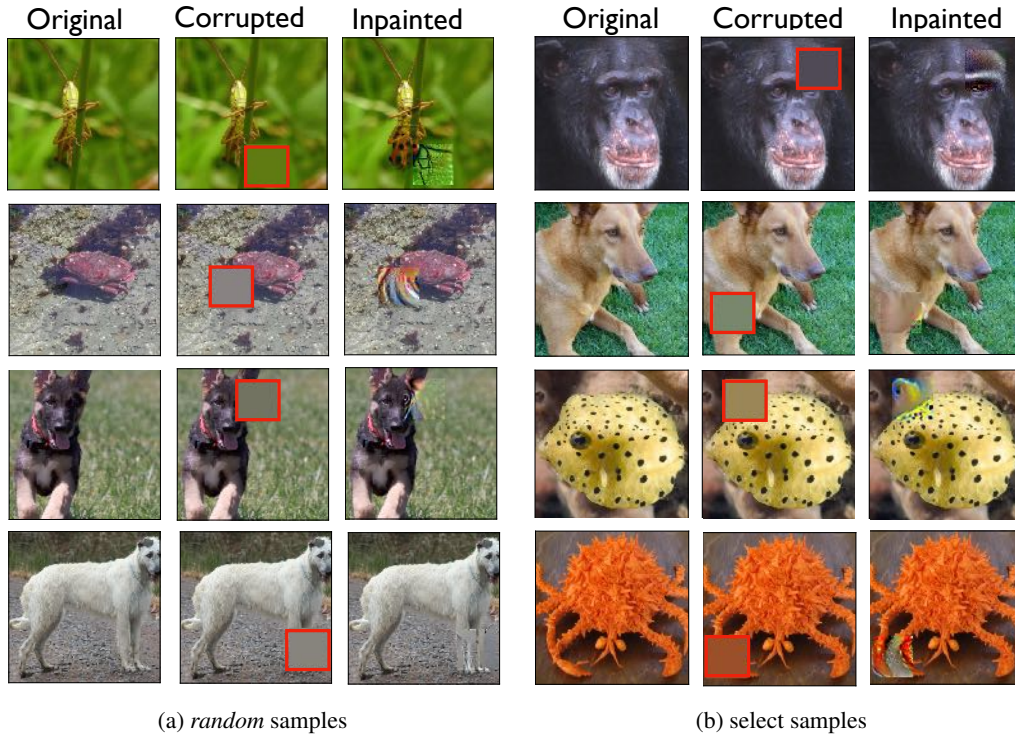

|       Original       Corrupted       Inpainted       Original       Corrupted       Inpainted       |
| (a) *random* samples | (b) select samples |

Figure 4: Image inpainting using robust models – *left:* original, *middle:* corrupted and *right:* inpainted samples. To recover missing regions, we use PGD to maximize the class score predicted for the image while penalizing changes to the uncorrupted regions.

In Figure 5, we provide sample translations produced by our approach using robust models—each trained only on the source and target domains for the Horse ↔ Zebra, Apple ↔ Orange, and Summer ↔ Winter datasets [Zhu+17] respectively. (For completeness, we present in Appendix B Figure 10 results corresponding to using a classifier trained on the complete ImageNet dataset.) In general, we find that this procedure yields meaningful translations by directly modifying characteristics of the image that are strongly tied to the corresponding domain (e.g., color, texture, stripes).

Note that, in order to manipulate such features, the model must have learned them in the first place— for example, we want models to distinguish between horses and zebras based on salient features such as stripes. For overly simple tasks, models might extract little salient information (e.g., by relying on backgrounds instead of objects[5]) in which case our approach would not lead to meaningful translations. Nevertheless, this not a fundamental barrier and can be addressed by training on richer, more challenging datasets. From this perspective, scaling to larger datasets (which can be difficult for state-of-the-art methods such as GANs) is actually easy and advantageous for our approach.

**Unpaired datasets.** Datasets for translation tasks often comprise source-target domain pairs [Iso+17]. For such datasets, the task can be straightforwardly cast into a supervised learning framework. In contrast, our method operates in the *unpaired* setting, where samples from the source and target domain are provided without an explicit pairing [Zhu+17]. This is due to the fact that our method only requires a classifier capable of distinguishing between the source and target domains.

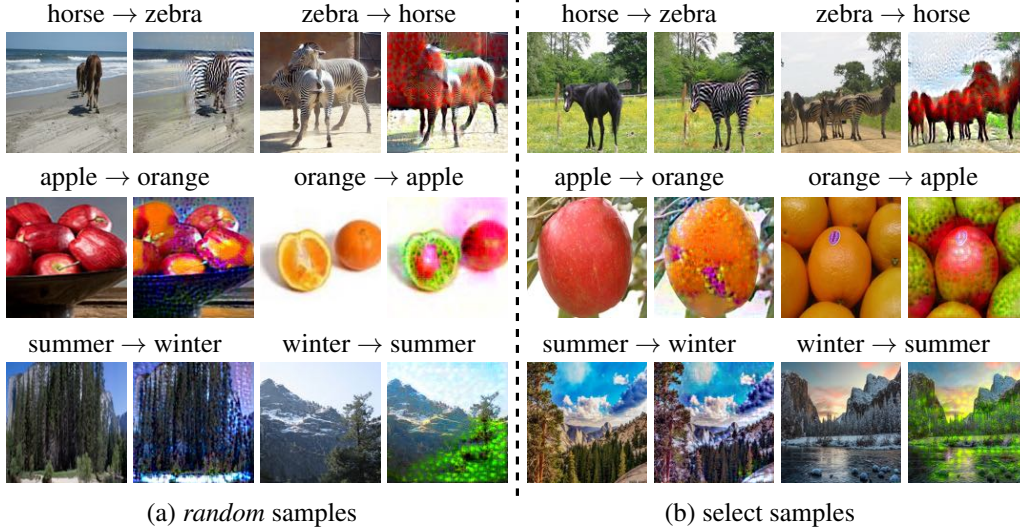

| (a) *random* samples | (b) select samples |

Figure 5: Image-to-image translation on the Horse ↔ Zebra, Apple ↔ Orange, and Summer ↔ Winter datasets [Zhu+17] using PGD on the input of an $\ell_2$-robust model trained on that dataset. See Appendix A for experimental details and Figure 9 for additional input-output pairs.

### 3.4  Super-Resolution

Super-resolution refers to the task of recovering high-resolution images given their low resolution version [DFE07; BSH12]. While this goal is underspecified, our aim is to produce a high-resolution image that is consistent with the input and plausible to a human.

In order to adapt our framework to this problem, we cast super-resolution as the task of accentuating the salient features of low-resolution images. This can be achieved by maximizing the score predicted by a robust classifier (trained on the original high-resolution dataset) for the underlying class. At the same time, to ensure that the structure and high-level content is preserved, we penalize large deviations from the original low-resolution image. Formally, given a robust classifier and a low-resolution image $x_L$ belonging to class $y$, we use PGD to solve

$$\hat{x}_H = \underset{||x'-\uparrow(x_L)||<\varepsilon}{\arg\min} \; \mathcal{L}(x',y) \tag{4}$$

where $\uparrow(\cdot)$ denotes the up-sampling operation based on nearest neighbors, and $\varepsilon$ is a small constant.

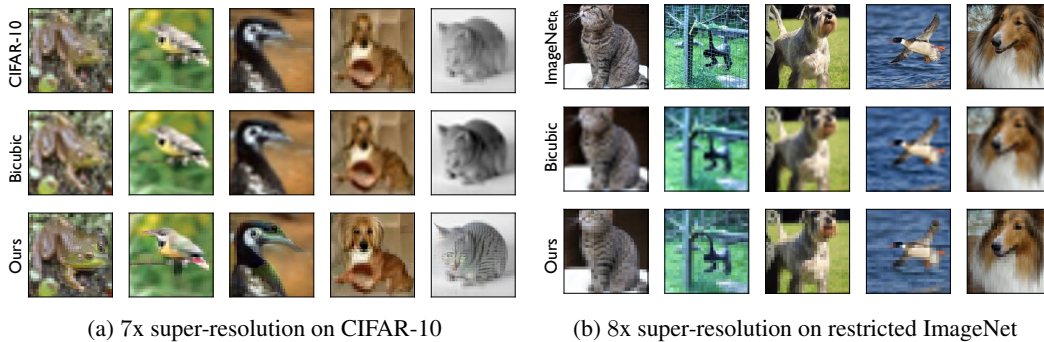

| (a) 7x super-resolution on CIFAR-10 | (b) 8x super-resolution on restricted ImageNet |

Figure 6: Comparing approaches for super-resolution. *Top: random* samples from the test set; *middle:* upsampling using bicubic interpolation; and *bottom:* super-resolution using robust models. We obtain semantically meaningful reconstructions that are especially sharp in regions that contain class-relevant information.

We use this approach to upsample *random* $32 \times 32$ CIFAR-10 images to full ImageNet size ($224 \times 224$)—cf. Figure 6a. For comparison, we also show upsampled images obtained from bicubic interpolation. In Figure 6b, we visualize the results for super-resolution on *random* 8-fold down-sampled images from the restricted ImageNet dataset. Since in the latter case we have access to ground truth high-resolution images (actual dataset samples), we can compute the Peak Signal-to-Noise Ratio (PSNR) of the reconstructions. Over the Restricted ImageNet test set, our approach yields a PSNR of 21.53 (95% CI [21.49, 21.58]) compared to 21.30 (95% CI [21.25, 21.35]) from bicubic interpolation. In general, our approach produces high-resolution samples that are substantially sharper, particularly in regions of the image that contain salient class information.

Note that the pixelation of the resulting images can be attributed to using a very crude upsampling of the original, low-resolution image as a starting point for our optimization. Combining this method with a more sophisticated initialization scheme (e.g., bicubic interpolation) is likely to yield better overall results.

### 3.5    Interactive Image Manipulation

Recent work has explored building deep learning–based interactive tools for image synthesis and manipulation. For example, GANs have been used to transform simple sketches [CH18; Par+19] into realistic images. In fact, recent work has pushed this one step further by building a tool that allows object-level composition of scenes using GANs [Bau+19]. In this section, we show how our framework can be used to enable similar artistic applications.

**Sketch-to-image.**    By performing PGD to maximize the probability of a chosen target class, we can use robust models to convert hand-drawn sketches to natural images. The resulting images (Figure 7) appear realistic and contain fine-grained characteristics of the corresponding class.

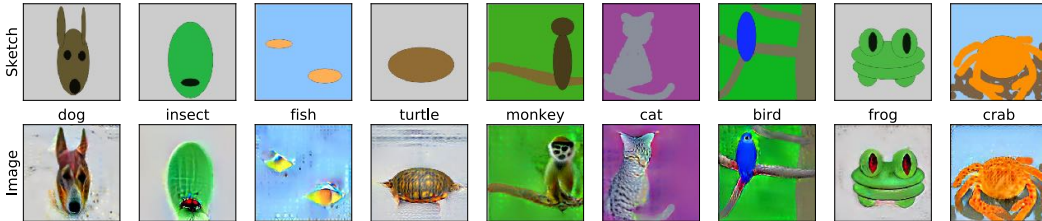

Figure 7: Sketch-to-image using robust model gradients. *Top:* manually drawn sketches of animals; and *bottom:* result of performing PGD towards a chosen class. The resulting images appear realistic looking while preserving key characteristics of the original sketches[6].

**Feature Painting.**    Generative model–based paint applications often allow the user to control more fine-grained features, as opposed to just the overall class. We now show that we can perform similar feature manipulation through a minor modification to our basic primitive of class score maximization. Our methodology is based on an observation of Engstrom et al. [Eng+19], wherein manipulating individual activations within representations[7] of a robust model actually results in consistent and meaningful changes to high-level image features (e.g., adding stripes to objects). We can thus build a tool to paint specific features onto images by maximizing individual activations directly, instead of just the class scores.

Concretely, given an image $x$, if we want to add a single feature corresponding to component $f$ of the representation vector $R(x)$ in the region corresponding to a binary mask $m$, we simply apply PGD to solve

$$x_I = \arg\max_{x'} R(x')_f - \lambda_P ||(x - x') \odot (1 - m)||. \tag{5}$$

In Figure 8, we demonstrate progressive addition of features at various levels of granularity (e.g., grass or sky) to selected regions of the input image. We can observe that such direct maximization of individual activations gives rise to a versatile paint tool.

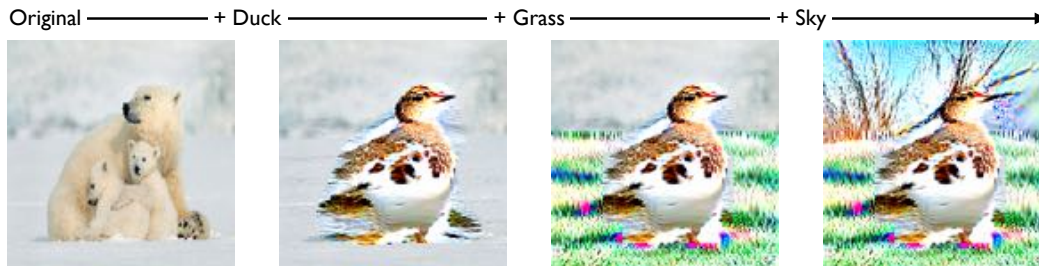

Figure 8: Paint-with-features using a robust model—we present a sequence of images obtained by successively adding specific features to select regions of the image by solving (5).

## 4    Discussion and Conclusions

In this work, we leverage the basic classification framework to perform a wide range of image synthesis tasks. In particular, we find that the features learned by a basic classifier are sufficient for *all* these tasks, provided this classifier is *adversarially robust*. We then show how this insight gives rise to a versatile toolkit that is simple, reliable, and straightforward to extend to other large-scale datasets. This is in stark contrast to state-of-the-art approaches [Goo+14; Kar+18; BDS19] which typically rely on architectural, algorithmic, and task-specific optimizations to succeed at scale [Sal+16; Das+18; Miy+18]. In fact, unlike these approaches, our methods actually *benefit* from scaling to more complex datasets—whenever the underlying classification task is rich and challenging, the classifier is likely to learn more fine-grained features.

We also note that throughout this work, we choose to employ the most minimal version of our toolkit. In particular, we refrain from using extensive tuning or task-specific optimizations. This is intended to demonstrate the potential of our core framework itself, rather than to exactly match/outperform the state of the art. We fully expect that better training methods, improved notions of robustness, and domain knowledge will yield even better results.

More broadly, our findings suggest that adversarial robustness might be a property that is desirable beyond security and reliability contexts. Robustness may, in fact, offer a path towards building a more human-aligned machine learning toolkit.

## Acknowledgements

We thank Chris Olah for helpful pointers to related work in class visualization.

Work supported in part by the NSF grants CCF-1553428, CCF-1563880, CNS-1413920, CNS-1815221, IIS-1447786, IIS-1607189, the Microsoft Corporation, the Intel Corporation, the MIT-IBM Watson AI Lab research grant, and an Analog Devices Fellowship.

## Footnotes

[3] Interestingly, the robust model used to generate these high-quality ImageNet samples is only 45% accurate, yet has a sufficiently rich representation to synthesize semantic features for 1000 classes.

[1]For ImageNet, there is a difference in resolution between BigGAN samples ($256 \times 256$), SAGAN ($128 \times 128$) and our approach ($224 \times 224$). BigGAN attains IS of 166.5. at $128 \times 128$ resolution.

[5]In fact, we encountered such an issue with $\ell_\infty$-robust classifiers for horses and zebras (Figure 11). Note that generative approaches also face similar issues, where the background is transformed instead of the objects [Zhu+17].

[6]Sketches were produced by a graduate student without any training in arts.

[7]We refer to the pre-final layer of a network as the representation layer. Then, the network prediction can simply be viewed as the output of a linear classifier on the representation.

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
