[Supplementary Material]

# A  Experimental Setup

## A.1  Datasets

For our experimental analysis, we use the CIFAR-10 [Kri09] and ImageNet [Rus+15] datasets. Since obtaining a robust classifier for the full ImageNet dataset is known to be a challenging and computationally expensive problem, we also conduct experiments on a "restricted" version if the ImageNet dataset with 9 super-classes shown in Table 2. For image translation we use the Horse $\leftrightarrow$ Zebra, Apple $\leftrightarrow$ Orange, and Summer $\leftrightarrow$ Winter datasets [Zhu+17].

| Class | Corresponding ImageNet Classes |
|---|---|
| "Dog" | 151 to 268 |
| "Cat" | 281 to 285 |
| "Frog" | 30 to 32 |
| "Turtle" | 33 to 37 |
| "Bird" | 80 to 100 |
| "Primate" | 365 to 382 |
| "Fish" | 389 to 397 |
| "Crab" | 118 to 121 |
| "Insect" | 300 to 319 |

Table 2: Classes used in the Restricted ImageNet model. The class ranges are inclusive.

## A.2  Models

We use the standard ResNet-50 architecture [He+16] for our adversarially trained classifiers on all datasets. Every model is trained with data augmentation, momentum of $0.9$ and weight decay of $5e^{-4}$. Other hyperparameters are provided in Tables 3 and 4.

| Dataset | Epochs | LR | Batch Size | LR Schedule |
|---|---|---|---|---|
| CIFAR-10 | 350 | 0.01 | 256 | Drop by 10 at epochs $\in [150, 250]$ |
| restricted ImageNet | 110 | 0.1 | 128 | Drop by 10 at epochs $\in [30, 60]$ |
| ImageNet | 110 | 0.1 | 256 | Drop by 10 at epochs $\in [100]$ |
| Horse $\leftrightarrow$ Zebra | 350 | 0.01 | 64 | Drop by 10 at epochs $\in [50, 100]$ |
| Apple $\leftrightarrow$ Orange | 350 | 0.01 | 64 | Drop by 10 at epochs $\in [50, 100]$ |
| Summer $\leftrightarrow$ Winter | 350 | 0.01 | 64 | Drop by 10 at epochs $\in [50, 100]$ |

Table 3: Standard hyperparameters for the models trained in the main paper.

## A.3  Adversarial training

In all our experiments, we train robust classifiers by employing the adversarial training methodology [Mad+18] with an $\ell_2$ perturbation set. The hyperparameters used for robust training of each of our models are provided in Table 4.

## A.4  Note on hyperparameter tuning

Note that we did not perform *any* hyperparameter tuning for the hyperparameters in Table 3 because of computational constraints. We use the relatively standard benchmark $\epsilon$ of 0.5 for CIFAR-10—the rest of the values of $\epsilon$ were chosen roughly by scaling this up by the appropriate constant (i.e. proportional to sqrt(d))—we note that the networks are not critically sensitive to these values of epsilon (e.g. a CIFAR-10 model trained with $\epsilon = 1.0$ gives almost the exact same results). Due to restrictions on compute we did not grid search over $\epsilon$, but finding a more direct manner in which to set $\epsilon$ (e.g. via a desired adversarial accuracy) is an interesting future direction.

| Dataset | $\epsilon$ | # steps | Step size |
|---|---|---|---|
| CIFAR-10 | 0.5 | 7 | 0.1 |
| restricted ImageNet | 3.5 | 7 | 0.1 |
| ImageNet | 3 | 7 | 0.5 |
| Horse $\leftrightarrow$ Zebra | 5 | 7 | 0.9 |
| Apple $\leftrightarrow$ Orange | 5 | 7 | 0.9 |
| Summer $\leftrightarrow$ Winter | 5 | 7 | 0.9 |

Table 4: Hyperparameters used for adversarial training.

## A.5 Targeted Attacks in Figure 2

| Dataset | $\epsilon$ | # steps | Step size |
|---|---|---|---|
| restricted ImageNet | 300 | 500 | 1 |

## A.6 Image-to-image translation

| Dataset | $\epsilon$ | # steps | Step size |
|---|---|---|---|
| ImageNet | 60 | 80 | 1 |
| Horse $\leftrightarrow$ Zebra | 60 | 80 | 0.5 |
| Apple $\leftrightarrow$ Orange | 60 | 80 | 0.5 |
| Summer $\leftrightarrow$ Winter | 60 | 80 | 0.5 |

## A.7 Generation

In order to compute the class conditional Gaussians for high resolution images ($224\times224\times3$) we downsample the images by a factor of 4 and upsample the resulting seed images with nearest neighbor interpolation.

| Dataset | $\epsilon$ | # steps | Step size |
|---|---|---|---|
| CIFAR-10 | 30 | 60 | 0.5 |
| restricted ImageNet | 40 | 60 | 1 |
| ImageNet | 40 | 60 | 1 |

### A.7.1 Inception Score

Inception score is computed based on 50k class-balanced samples from each dataset using code provided in `https://github.com/ajbrock/BigGAN-PyTorch`.

## A.8 Inpainting

To create a corrupted image, we select a patch of a given size at a random location in the image. We reset all pixel values in the patch to be the average pixel value over the entire image (per channel).

| Dataset | patch size | $\epsilon$ | # steps | Step size |
|---|---|---|---|---|
| restricted ImageNet | 60 | 21 | 0.1 | 720 |

## A.9 Super-resolution

| Dataset | ↑ factor | $\epsilon$ | # steps | Step size |
|---|---|---|---|---|
| CIFAR-10 | 7 | 15 | 1 | 50 |
| restricted ImageNet | 8 | 8 | 1 | 40 |

# B   Omitted Figures

Horse ↔ Zebra

Apple ↔ Orange

Summer ↔ Winter

Figure 9: Random samples for image-to-image translation on the Horse ↔ Zebra, Apple ↔ Orange, and Summer ↔ Winter datasets [Zhu+17]. Details in Appendix A.

Horse → Zebra                    Apple → Orange

Figure 10: Random samples for image-to-image translation on the Horse ↔ Zebra and Apple ↔ Orange datasets [Zhu+17] using the *same* robust model trained on the *entire ImageNet* dataset. Here we use ImageNet classes "zebra" (340) and "orange" (950).

Figure 11: Training an $\ell_\infty$-robust model on the Horse $\leftrightarrow$ Zebra dataset does not lead to plausible image-to-image translation. The model appears to associate "horse" with "blue sky" in which case the zebra to horse translation does not behave as expected.

Figure 12: Failure cases for image inpainting using robust models – *top:* original, *middle:* corrupted and *bottom:* inpainted samples. To recover missing regions, we use PGD to maximise the class score of the image under a robust model while penalizing changes to the uncorrupted regions. The failure modes can be categorized into "good" failures – where the infilled region is semantically consistent with the rest of the image but differs from the original; and "bad" failures – where the inpainting is clearly erroneous to a human.

Figure 13: Random samples generated for the CIFAR dataset.

Figure 14: Random samples generated for the Restricted ImageNet dataset.

Figure 15: Random samples generated for the ImageNet dataset.

Figure 16: Random samples from a random class subset.

CIFAR10

Restricted ImageNet

ImageNet

Figure 17: Samples from class-conditional multivariate normal distributions used as a seed for the generation process.