[Reviews · NeurIPS 2019]

Reviewer 1



I quite enjoyed reading this paper. I do want the paper to include a brief discussion about the difficulties of training robust classifiers on large datasets. Are there any schemes to mitigate these difficulties? If such difficulties persist then the purposed approach would not really replace our reliance upon GANs. GANs are tedious to train, but at least these work really well. For image generation and image-to-image translation: I noticed that a number of generated images are very noisy. Some of the images seem blended from multiple images. Is it a deficiency in the method? Is it due to lack of training? Or is due to a lack of access to data? For inpainting: I assume that your method can work in situations where multiple non-overlapping regions are missing? I think that the proposed approach has a lot of promise. Some of the figures presented in the paper, however, doesn't do a good job of convincing a reader that the proposed method will eventually beat the performance achieved via using GANs.

Reviewer 2



Summary: the submission shows that a robustly trained classifier encodes strong visual priors about the world. This is a result of Tsipras [37] method and findings, and well described in L42-43, L47-48, and Figure 2. Hence they were able to show meaningful results for a wide set of class-conditioned image synthesis tasks. Being robustly trained (i.e. using eq 2 rather than eq 1) is indeed crucial for achieving such results. Elaborated Strengths: S1) the main point of the submission (a robust classifier encoding strong priors about the visual world) is valid. I found it insightful and would happily read a paper such as this one that would analysis/proves this. Here the main point is proved via experimentally showing the robust classifier could solve a various conditioned image synthesis tasks. Elaborated Weaknesses W1) The submission in numerous parts claims to be solving "computer vision" or a big chunk of it and pitches the method as a computer vision "toolkit". Eg in the title, L30 "we..build a versatile computer vision toolkit", conclusion "framework can be leveraged to perform a wide range of computer vision tasks", etc. This claim is both technically false and unnecessary. First, to be precise, the method can handle and is demonstrated to do *image synthesis* and in a *class-conditioned* manner. This does not mean "computer vision". This is indeed more similar to computer graphics objective (synthesis), and leaves out most of computer vision which is about inferring low dimensional abstractions out of visual observations (the inverse). Does this method have anything to provide about object detection, camera pose estimation, surface normal prediction, 3D reconstruction, etc? These are examples of common vision tasks and have no synthesis aspect to them. Second, this is just an unnecessary claim and a distraction, in my opinion. Just concretely showing that, as this submission does, a robust classifier encodes strong visual priors about the world is a good enough finding. I would simply state that in a straightforward manner, without unsubstantiated extrapolations to the bigger umbrellas. This is a signifiant issue in the current submission, and I believe it is crucial to majorly revise the introduction, conclusion, and title. My preliminary rating is based on the assumption that this can be sufficiently addressed in the camera ready. Please comment on this in the rebuttal. W2) The method in its current form is limited to tasks that are about image synthesis and are class-conditioned. This should be made clear upfront. W3) Suggestion: I see a missed opportunity here. The paper focuses on image synthesis tasks. If the robust classifier indeed encodes better visual priors about the world, I would expect its embedding to be a better and more robust *feature* as well. For instance, if you compare the embedding of a network pre-trained on imagenet as an off-the-shelf vision feature (as it is commonly done), would the feature be better if the network was trained with the robust objective? This can be tested by measuring the transfer learning performance of the feature in a style similar to taskonomy CVPR18 analysis. While this is not a basis for rejection, I would recommend that the authors consider doing it for strengthening the submission and broadening the impact. Even if the results dont end up supportive, I would include it as negative results and an interesting point of discussion for the community, as I would find the result intriguing either way. More comments: C1: as L112 states, it should be possible to do the inpainting (and some of the other tasks) without knowledge of the target class. Have you attempted that, e.g. by adding an argmax for the target class to the synthesis objective? Similarly, how much does knowledge of the correct target class impact the synthesis results? Eg how would the inpainting results in fig 4 look like if other target classes were used for the same image? Could be an interesting visualization/analysis. C2: for the sake of completeness, I would suggest in some of the visualizations you add the corresponding results achieved by a non robust network. We know from the adversarial literature that the results would look poor, but could be good to include an example in some of the figures as that's the main point of this submission. C3: both super resolution and inpainting objectives (ie eq 3 & 4) have a component in them in which part of the input image is wished to be preserved. In eq 3 that is implemented using an additional term in the loss while in eq 4 that is implemented as a constraint over x' while it could be implemented as a loss term as in eq 3 too. Was this switch deliberate? If so, any insights why? C4: in feature painting and eq 5: how do you know what semantics each component of R corresponds to? How exactly the R vector is formed here? Also, how do you get the masks 'm' for proper feature painting? C5: have you considered doing the same, but using networks that are trained for tasks other than object classification, eg single image 3D tasks such as surface normal prediction? I suspect the visual primitives they encode would be substantially different from and complementary to imagenet based ones. Could be an interesting study. C6: can you pose your good synthesis results and general concept wrt to the works that show the inductive bias in the neural network architectures, even in a random network, results in good synthesis output? (e.g. Deep Image Prior [38])

Reviewer 3



This paper tries to leverage the generative behavior of classifiers to perform a range of image processing tasks, including image generation, inpainting, translation, super-resolution. I like this paper and I think it's a super fun read. That being said, I have some reservations about publishing it. Here's my major criticisms: 1) The whole thesis of this paper seems to be that *robustness* is the key to unlocking the generative behavior needed to perform these tasks. However, no comparisons are made with non-robust models, and I find this odd. My experience is that even non-robust models can exhibit generative behavior, and I'm curious what would happen if non-robust models were used. This is especially interesting for problems like ImageNet where even the best adversarially trained models aren't very robust. Can you really observe a difference between robust and non-robust in this case? For perspective, this well-known distill article does visualization on NON-robust networks, and gets (in many ways) even better results than those presented in the paper under review: https://distill.pub/2017/feature-visualization/ The paper also cites a long list of articles that optimize images to maximize the activation of an output neuron (just like the paper under review) with often spectacular results. Not only do these papers have huge conceptual overlap with the paper under review, but they do it all without adversarial training. 2) This paper has a large amount of conceptual overlap with the Tsipras paper. The most impressive results in the paper under review are for image generation, but these experiments are nearly identical to what was done in the Tsipras paper (although with a wider range of datasets). The strong overlap with (https://distill.pub/2017/feature-visualization/) and the many citations therein is also concerning. 3) Other similar variational methods that also solve inpainting and super-resolution problems are not discussed or compared to. What about the "deep image prior"? Plug-and-play methods? Or "regularization by denoising"? 4) There are no real surprises in this paper. By and large, the results presented are pretty much what I'd expect: these simple optimization methods can perform basic imaging tasks, but can't compete with (or even come close to) the state of the art because of the artifacts that arise from adversarial behavior. I also have one more minor criticism: the performance of these methods is not good. Normally I'd be ok with this, since I like to see exploratory ideas rather than just engineering to achieve high benchmarks (which is why I list this as a minor rather than a major criticism). However, in this case, it feels like the authors are making a lot of unjustified claims that their methods work great. If I just read the text of the paper without looking at figures, I'd have the impression that these methods compete with state of the art performance. Claims that the produced images are "realistic" are a bit bombastic. Furthermore, bragging about how the method achieves higher inception scores than BigGAN is strange when it can't even scratch the surface of what BigGAN does. I take the results here to be a confirmation that inception score is a meaningless metric, rather than a confirmation that robust models produce good results. I find it odd that the paper never seems to acknowledge the gap between this approach and other generative methods, or other variational methods. Finally, the following is a curiosity rather than a criticism: I can see a lot of pixelation in the super-resolution results. This clearly arises because of the nearest-neighbor interpolation that is used as a base image. What happens if you just treat this like an inpainting or deblurring problem, which is the approach more commonly taken in the super-resolution literature? Overall this is a fun paper, and the sketch-to-image stuff is clever. I'm not sure I'd prioritize this paper over other good papers for the reasons describes above. Despite the fun, there's not a lot of novelty in it. Image generation has been done before by lots of other authors, including methods that maximize class labels as is done here. The main contribution here is doing it with robust nets rather than clean trained nets. I'm not sure how significant this is though, especially since the generative behaviors the authors claim are unique to robust models have been observed many times before on non-robust models, and the authors present no comparisons to similar variational methods methods or non-robust nets. Update: I read the rebuttal and I'm sympathetic to some of the arguments. I see how the authors think that their results are better than those in the links they provided. However the authors used a specialized initialization (a Gaussian with a specially chosen covariance) rather than an iid random initialization, and may have a number of other differences in their setup. I'm confident the specialized initialization the authors used made an impact on the quality of their results, otherwise they wouldn't have used it. How would this initializer impact the quality of results from a non-robust network? I don't know. The above comments aren't means to doubt the superiority of a robust network approach, nor am I assuming that the initializer would bring a non-robust network result up to the quality of the robust network result. My major concern with the paper is that I shouldn't have to speculate on these issues. Most papers fall into the category of either a theory paper or an experiments paper. For an experiments paper, I expect there to be ablation studies (where appropriate), and comparisons to the state of the art (where appropriate). How do your methods compare when I plug in a non-robust network? Is there an ablation study to show in impact of the initialization? What if you did inpainting using a standard loss function? How does this compare to another variational inpainting method, like plug and play or RED? The first question I ask above still seems somewhat glaring. Given how prominent the claims in the paper are about the importance of using a robust classifier (it's mentioned in the title), this issue should be addressed. Finally - I like the new title you suggested. This is really a paper about image synthesis and not "computer vision."

[Author Response · NeurIPS 2019]

We thank the reviewers for their detailed feedback—it will help improve our paper.

**(R1) Training difficulties.** Training the models was actually quite simple (involved adding only 5-6 lines of code on
top of standard/boilerplate model training code), and extremely consistent and stable. In fact, our method scales easily
to larger datasets (unlike GANs) and does even better on "harder", more fine-grained tasks.

**(R3 W1, R3 W2)** The reviewer's concern about overclaiming is well received (and appreciated). We are very much
aware that our approach does not "solve CV" and did not intent to claim that. In fact, that's why we explicitly list the
tasks we perform in the introduction (and our intention was to use the phrase "CV toolkit" in the same way one might
describe GANs as enabling a "CV toolkit", despite not being all-powerful).

However, in the light of this feedback, we decided to make this point more clear. To this end, we will change our title
to "Image Synthesis with a Single (Robust) Classifier". Similarly, we will modify the abstract, intro, and conclusion
accordingly, changing "CV toolkit" to "toolkit for (class-conditioned) image synthesis tasks," etc., as suggested.

**(R3, R6) Quality of synthesized images.** We want to emphasize that the goal of our paper is not to directly improve
on state-of-the-art for any of the tasks that we consider. Instead, our goal is to introduce robust classifiers as a new and
promising framework that even "out of the box" (i.e., without any real engineering) is able to perform these tasks at
a reasonable level. Crucially, our framework does not seem to have hit any fundamental obstacles preventing it from
attaining state-of-the-art results. To contrast it with GAN-based approaches, one should note that:

**(A)** The "out-of-the-box" GAN framework (minimizing the theoretically motivated loss with standard architectures)
gives results that are far from satisfactory. In particular, they are worse than what we get with an off-the-shelf ro-
bust classifier and no optimizations/regularization/proprietary datasets in Fig. 3. Attaining the current state-of-the-art
results with GANs required years of effort that involved devising computational and computer vision-based optimiza-
tions. In contrast, our ImageNet results (Fig. 3) would be state-of-the-art even 3-4 years after GANs were introduced.
Indeed, scaling GANs to ImageNet was possible only recently—whereas robust classifiers actually seem to *improve*
with more fine-grained classification tasks. **(B)** The training dynamics of GANs are complex, unstable, and hard to
interpret. In our case, the dynamics and the learned representations are just fairly natural variants of the standard (and
rather well-understood) discriminative framework.

Finally, we find the R6's comment on "bragging" about the Inception Score somewhat unfair. Our paper explicitly
discusses why I.S. is not a good metric to compare the two approaches, and just points out that it is unreasonably high.

**Non-robust models work/Unsupprising/Novelty over distill.pub.** Overall, we are surprised by these comments. We
are not aware of prior work that performs all these vision tasks with an off-the-shelf discriminative model without task-
specific optimization. In fact, even the work the reviewer cites `https://distill.pub/2017/feature-visualization/`
`appendix/` notes:"In this layer [the final layer, used for feature painting] visualizations become mostly nonsensical
collages...*neurons do not seem to correspond to particularly meaningful semantic ideas anymore.* [emph. added]."

Indeed, replicating our experiments using a standard network (without regularization) fails completely (this has been
observed by several papers, including the linked Feature Visualization one—we will include examples in our ap-
pendix). Even when one employs quite intricate regularization methods (gradual upscaling, gradient blur, blur re-
duction, random shifts, etc.) the generated images from a single discriminative model are qualitatively worse than
what we present. For example, compare `https://bit.ly/2JYr0yh` and `https://arxiv.org/abs/1507.02379`,
(which are the most recent class visualization sans trained GAN/AE we found, via the distill.pub article), and even the
image labeled "Class Logits" in the distill.pub paper, to Figure 3 in our work.

Finally, as mentioned earlier, prior work relies heavily on additional regularization—our work highlights that when
using robust models none of this is necessary (and in fact, the promising advancements in regularization for non-robust
networks can even be integrated with robust classifiers in the future to get even better results).

**(R6) Overlap with Tsipras et al.** While Tsipras et al was a motivation for this work, we disagree that there exists
a significant overlap between the two works. In particular, the Tsipras et al. paper simply shows that untargeted
attacks on robust models, starting from test set images, seem to change relevant features in the input. In our work,
we show that one can actually leverage robust classifiers for diverse class-conditional image generation, inpainting,
superresolution, etc. which as far as we are aware have never been done with a standard discriminative architecture.

**Minor comments (space constrained): (R1)** Our method is general and can work with arbitrary pixel masks. We will
show this in revised appendix along with **(R3 C2)** our results for standard networks (which are indeed poor). **(R3 C3)**
The difference is purely for implementation convenience, as the constrained and regularized version are technically
equivalent (via Lagrange multiplier property) **(R3 C6)** We view robustness as a prior that allows for good synthesis
results. In that sense, it is complementary to other priors, e.g., the deep convolutional prior.

[Meta-Review · NeurIPS 2019]

The submission is an empirical work which shows that a robust classifier can be used for various image synthesis tasks. The empirical study is solid, extensive, and conclusive. Both quantitative and qualitative results suggest that the hypothesis is correct and robust classifiers solve various image synthesis tasks with a simple likelihood maximization. The submission is interesting and would be a good addition to the conference. One of the issues raised by the reviewers is additional experiments specifically trying the proposed methodology with a non-robust network. Including it in the camera-ready version would be good for the sake of completeness. The paper has a potentially large impact on the community. Synthesizing images requires learning the data distribution of natural images. And, the paper suggests that the robust networks do a better job in learning this prior. This result might have a large impact on the community since the learned data distribution can be used in various downstream tasks.